# Rheumatoid Arthritis: Pathogenic Roles of Diverse Immune Cells

**DOI:** 10.3390/ijms23020905

**Published:** 2022-01-14

**Authors:** Sunhee Jang, Eui-Jong Kwon, Jennifer Jooha Lee

**Affiliations:** 1Division of Rheumatology, Department of Internal Medicine, Seoul St. Mary’s Hospital, College of Medicine, The Catholic University of Korea, Seoul 06591, Korea; sunshinerosa@naver.com (S.J.); ejkwon@catholic.ac.kr (E.-J.K.); 2Yonsei Hangang Hospital, 25 Mapodaero, Mapogu, Seoul 04167, Korea; 3Chemical, Biological, Radiological, and Nuclear (CBRN) Defense Research Institute, Armed Forces CBRN Defense Command, Seoul 06591, Korea

**Keywords:** rheumatoid arthritis, epidemiology, diagnosis, pathogenesis, autoantibodies, precision medicine

## Abstract

Rheumatoid arthritis (RA) is a chronic, systemic autoimmune disease associated with synovial tissue proliferation, pannus formation, cartilage destruction, and systemic complications. Currently, advanced understandings of the pathologic mechanisms of autoreactive CD4+ T cells, B cells, macrophages, inflammatory cytokines, chemokines, and autoantibodies that cause RA have been achieved, despite the fact that much remains to be elucidated. This review provides an updated pathogenesis of RA which will unveil novel therapeutic targets.

## 1. Introduction

Rheumatoid arthritis (RA) is a chronic, inflammatory, systemic autoimmune disease that is associated with progressive disability, systemic complications, and early death [1,2]. RA is characterized by synovial inflammation and hyperplasia, production of autoantibodies including rheumatoid factor (RF) and anti-citrullinated protein antibody (ACPA), cartilage and bone deformities, and systemic features including cardiovascular, pulmonary, psychological, skin, and skeletal disorders [2].

In recent decades, we have obtained new genetic and pathogenetic insights along with new developments in RA disease assessment and therapeutic strategies, which have led to the approval of a variety of novel therapies [3]. In this review, we focus on the roles of diverse immune cells along with the wide spectrum of molecular mechanisms involved in the pathogenesis and clinical expression of RA, as well as their possible contribution to treatment response and precision medicine.

## 2. Epidemiology

Most epidemiological studies in RA have been conducted in Western countries, showing an RA prevalence in the range of 0.5–1.0% in the US [4]. In general, women are 2–3 times more likely to develop RA than men. Indeed, the cumulative lifetime risk of developing adult-onset RA has been roughly estimated at 3.6% for women and 1.7% for men [5,6].

RA has a strong genetic component. Twin studies have estimated the heritability of RA to be approximately 60% [7]. This number is observed in ACPA-positive patients, while estimates of seronegative diseases are lower. However, the disease concordance of identical twins is only 12–15%, indicating that environmental factors also play an important role in susceptibility.

About 100 loci have been identified across genomes harboring RA susceptibility variants by genome-wide association studies [8,9,10], with fine mapping [11], candidate gene approaches [12,13], and a meta-analysis of genome-wide association studies involving >100,000 individuals [14]. In particular, specific class II human leukocyte antigen (HLA; also known as major histocompatibility complex (MHC)) loci, which encode MHC molecules that may contain a shared epitope, show a very strong susceptibility to RA, consistent with classical findings [15].

Smoking, silica exposure, and periodontal disease are environmental risk factors for developing RA [16,17,18].

Both genetic and environmental risk factors contribute to RA, and multiple risk factors may be required before the threshold at which RA is triggered. Disease progression includes asymptomatic synovitis and the initiation and dissemination of autoimmunity against altered auto-proteins that can occur years before clinical symptoms begin [3].

## 3. Diagnosis

The diagnosis of RA is based on the 2010 American College of Rheumatology (ACR)/European League Against Rheumatism (EULAR) classification criteria (Table 1) [19]. Application of these criteria provides a score of 0–10, with a score of ≥6 being satisfactory for the diagnosis of definite RA. The 2010 ACR/EULAR criteria included serologic testing (RF or ACPA). The diagnostic criteria for ACPA are presentation of an early disease course and prediction of an aggressive disease course [20].

In addition to RF and ACPA, antibodies to mutant citrullinated vimentin (MCV) may be useful additional biomarkers in an array of diagnostic tools for RA. Anti-MCV antibodies recognize a protein that is derived from apoptotic macrophages and is present in the synovium of RA patients [21]. A meta-analysis showed that anti-MCV antibodies demonstrate comparable diagnostic value to anti-CCP and RF, and they can be an effective diagnostic marker for RA. Thus, anti-MCV antibodies may be an alternative test, used in patients suspected of RA, but with anti-CCP and RF negative [22].

## 4. Molecular Mechanisms in the Pathogenesis

### 4.1. Synovium

Synovial tissue in RA patients can be considered as tertiary lymphoid tissue (TLT) or an ectopic lymphoid structure (ELS). Its structure resembles the secondary lymphoid tissue where T cell and B cell differentiation occurs. TLTs are correlated with autoantibody titers, inflammatory cytokine levels, and disease severity in RA patients, indicating that TLTs are related to the persistent inflammation in RA [23].

There are two important pathogenetic changes in the synovial membrane of RA. Regarding the first, the intima is greatly expanded due to the increase in and activation of both synoviocyte types—macrophage-like synoviocytes (MLSs) and fibroblast-like synoviocytes (FLSs), which are prominent sources of cytokines and proteases, including integrins, selectins, and members of the immunoglobulin superfamily. MLSs produce a variety of pro-inflammatory cytokines, including IL-1, IL-6, and tumor necrosis factor (TNF)-α [3]. FLSs express not only IL-6, but also huge amounts of matrix metalloproteinases (MMPs) and small-molecule mediators (prostaglandins and leukotrienes) [24]. In addition, FLSs aid in the activation of immune responses by interacting with immune cells and by supporting ELS formation in synovial tissues [25]. The second is infiltration of adaptive immune cells into the synovial sublining. This leads to hallmark “pannus” formation at cartilage–bone interfaces [26]. Pannus can be composed of macrophages, FLSs, dendritic or plasma cells, and mast cells, and it mediates damage and erosion formation in later disease [27,28].

Half of the sublining cells are CD4+ memory T cells, which either diffusely infiltrate tissues or form ectopic germ centers where mature B cells proliferate, differentiate, and produce antibodies [3]. There are also B cells, plasmablasts, and plasma cells present, many of which produce RF or ACPAs [29].

### 4.2. T Cell-Mediated Immune Response in RA

The long-standing association of the human leukocyte antigen (HLA)-DRB1 locus with patients with RA suggests the influence of T cell selection and antigen presentation in the induction of autoreactive immune responses [2,30]. RA is driven by CD4+ T lymphocytes, meaning IL-6 is an important mediator of bony destruction in RA because it regulates T lymphocyte production and inflammation [31]. Although IFN-γ levels are not high in the synovial membrane of patients with RA, the cytokine is regarded pivotal in RA pathogenesis. Not only IFN-γ-producing Th1 cells but also IL-17-producing helper T (Th17) cells have important roles in RA development [30,32]. There are two types of Th17 cells, “pathogenic” Th17 cells and “non-pathogenic” Th17 cells, depending on the cytokine milieu present during the differentiation process. Usually, “pathogenic” Th17 cells are considered to be positive regulators of immune responses because they produce pro-inflammatory cytokines, including IL-17A, IL-17F, and IL-22. Conversely, “non-pathogenic” Th17 cells may secrete immunosuppressive factors such as IL-10 to negatively modulate the immune response [33]. In RA, pathogenic Th17 cells play a very important role. Among different cytokines that this Th17 subtype produces is granulocyte-macrophage colony-stimulating factor (GM-CSF), or IL-22. Some studies have shown that the signaling pathway of IL-22 (GM-CSF) may be activated in RA pathogenesis [34,35].

IL-17 induces pro-inflammatory cytokines such as TNF-α, IL-1β, and IL-6 in the cartilage, synovial cells, macrophages, and bone cells [36,37]. IL-17 also stimulates the production of several chemokines, including CXCL1 (KC/Groα), CXCL2 (MIP2α/Groβ), CXCL8(IL-8), CCL2 (MCP1), CCL7 (MCP3), and CCL20 (MIP-3α). They serve to enhance inflammation by recruiting neutrophils, macrophages, and lymphocytes to the synovial membrane [37]. IL-17 deficiency ameliorated the development of arthritis [38,39]. Therefore, several IL-17A blockers have been evaluated in clinical trials, including the anti-IL-17A monoclonal antibodies (secukinumab and ixekizumab) and the anti-IL-17 receptor subunit A monoclonal antibody (brodalumab). However, the results showed no incremental benefit in patients with background methotrexate who have had an inadequate response to a prior TNF-α inhibitor [40,41]. Not only IL-17 but also IL-22 promotes inflammatory responses in RA synovium by inducing proliferation and chemokine production of synovial fibroblasts [42]. In RA, pro-inflammatory mediators may interfere with T cell regulation and induce T cell plasticity. The inflammatory environment also induces regulatory (Treg) cell expansion so that large numbers of proliferating Treg cells are detected in the patient’s inflamed joints [43]. Treg cells may become less functional or even pathogenic in autoimmune inflammatory environments. For example, IL-1β and IL-6 downregulate forkhead box P3 (Foxp3) expression, and they reduce Treg inhibitory function [30,44]. Foxp3 is a key regulator or immunosuppressor in Treg cells and participates in the gene expression, function, and survival of Treg cells. Its expression is regulated by transcriptional regulation, epigenetic regulation, and post-translational regulation and is essential for the maintenance of immune self-tolerance [45,46]. The transcription factors Nuclear factor of activated T-cells (NFAT), signal transducer and activator of transcription 5 (STAT5), and Foxo1 regulate the expression of Foxp3 by directly interacting with the Foxp3 gene promoter. Elements of the conserved noncoding sequence (CNS) in the Foxp3 gene region recruit transcription factors to regulate gene expression [47,48].

In summary, T cells (T helper (Th)1, Th17, Treg, and Th22) initiate a serial cascade (rolling, arrest, spreading, crawling, and migration) and eventually extravasate from blood vessels to the inflamed joint [30]. Recent studies have shown that differentiated CD4+ T cell subpopulations exhibit a high degree of plasticity. That is, FoxP3+ regulatory T cells (Tregs) and Th1 and Th17 effector T cells demonstrate high levels of plasticity, allowing functional adaptation to various physiological situations during the immune response via IL-1, IL-6, IL-12, IL-17, and IFN-γ. Therefore, the plasticity of CD-4+ T cells may have evolved to remain resilient, as well as stable, making the immune system most flexible to respond to pathogens and environmental changes. However, this flexibility also includes potential threats to the host. This is because deregulation of this system increases the risk of autoimmune development. Therefore, factors regulating Treg and Th17 plasticity could be goals for immunotherapy targeting the manipulation of the immune system in the setting of autoimmune diseases [49].

Despite the abundance of T cells in the synovial environment, the functional role of T cells remains not yet fully understood. Direct targeting of T cells by cyclosporine or T cell-depleting therapeutics has been shown to be limited or ineffective [50]. On the other hand, abatacept, an inhibitor of CD28-mediated T cell activation, has been shown to be effective in regulating inflammation during RA. Abatacept’s efficiency in achieving remittance in RA can be attributed, in part, to its ability to enhance immune regulatory cells, particularly IL-35 + IL-10 + regulatory B (Breg) cells [51].

### 4.3. B Cell-Mediated Immune Response in RA

The functions of B cells, namely, autoantibody production, antigen presentation, and cytokine secretion, are related to the pathogenesis of RA (Figure 1) [23,52].

#### 4.3.1. Autoantibody Production

Autoantibodies are mainly secreted and produced by Toll-like receptor (TLR) driven autoreactive B cells after differentiation into plasma cells [53]. The cross-reactivity of some proteins with post-translational modification (PTM) and foreign antigens may lead to the expansion of autoreactive B cells in RA [54]. Additionally, excessive activation-induced cytidine deaminase (AID) expression in B cells of RA patients associates with the high levels of T helper cell cytokines IFN-γ and IL-17, leading to the development of anti-CCP and RF [55]. Proliferation of germinal center B cells replaces the naive circulating B cells that currently form the follicular mantle region. The germinal center has two zones: the dark zone (filled with dividing GC B cells known as centroblasts), and the light zone (filled with follicular dendritic cells (FDCs) with surface-bound antigens). FDCs bind antigens in the form of immune complexes. Dividing centroblasts acquire somatic hypermutations in immunoglobulin variable region genes [56]. The autoantibodies of RA mainly include RF, ACPA, anti-modified citrullinated vimentin antibody, anti-carbamylated protein antibody, anti-PAD-4 antibody, and anti-GPI antibody. Among the RA-related autoantibodies, ACPA has the most remarkable prognostic value concerning RA onset among symptomatic at-risk patients [57]. There are some differences between RF and ACPA. ACPA-positive B cells undergo multiple rounds of germinal center responses that lead to a high level of somatic hypermutations and isotype switching. In contrast, RF-positive B cells undergo several rounds of germinal center responses with only a modest number of somatic mutations including two transcription factors (BACH2 and SOX11) and can be activated by innate immune mechanisms, while ACPA-positive B cells show enriched differentially expressed genes associated with T cell-dependent B cell differentiation. Therefore, the citrulline-specific immune response produces relatively stable long-living plasma cells and ACPA autoantibodies, whereas the RF response is characterized by the generation of short-living plasma cells and fluctuating RF levels (Figure 2) [58].

Rheumatoid factor (RF) is an antibody that recognizes the Fc portion of immunoglobulin-G (IgG). Additionally, RF was the first type of autoantibody detected in RA and used in the 1987 ACR classification criteria for RA [59]. Although the classical Waaler–Rose assay relies mainly on IgM antibodies, RF activity can be found in almost all types of immunoglobulins (IgA, IgG, and IgM) [60]. Various titers and isotypes of RF can be observed in a wide range of different diseases [61], and high titers of IgM and IgA are thought to be highly indicative of RA [62]. Whether RF levels correlate with clinical disease activity is in debate. RF levels have the potential to revert and convert during the early course of disease. Additionally, RF fluctuations are not associated with clinical outcomes [63]. Nevertheless, RF remains a useful diagnostic marker of RA used in routine clinical practice [64].

Anti-citrullinated protein antibodies (ACPAs) are an important parameter to help rheumatologists set a diagnosis of early RA and start initial treatment [65]. Since inflammation plays a central role in the pathogenesis of RA, it has been suggested and demonstrated that ACPA can activate immune cells and upregulate pro-inflammatory cytokine production [64].

Multiple studies have demonstrated that elevated ACPA levels are present in preclinical RA, and the presence of ACPA is highly specific for predicting the future development of RA, with a specificity of 85–95% and a sensitivity of 67% [66,67,68]. Therefore, ACPAs have been included in the widely used ACR/EULAR 2010 classification criteria [20]. ACPAs are directed against citrulline residues on proteins or peptides. Citrullination is an irreversible PTM of arginine mediated by enzymes called peptidyl arginine deaminases (PADs) [64]. Consequently, many citrullinated proteins (fibrinogen, a-enolase, vimentin, and collagen type II) have been shown to be recognized by ACPA [69,70,71,72]. In mouse models, murine and passively transferred human ACPAs substantially contributed to arthritis [73]. In vitro studies have shown that ACPA exerts a biological function, particularly by binding to Fc receptors expressed by immune cells of the myeloid lineage and activating the complement system through classical and alternative pathways [74]. It was demonstrated that complexes consisting of citrullinated fibrinogen and ACPA (CitFibr-ACPA) present in the RA synovial membrane can stimulate macrophages via dual engagement of TLR-4 and FcγR, resulting in the synergistic induction of TNF-α production. This suggests a potential role for citrullination in increasing the efficacy of endogenous innate immune ligands and providing insight into the mechanisms by which anti-citrulline autoimmunity may contribute to the pathogenesis and spread of inflammation in RA [75,76,77]. In particular, the strong FcR-mediated or complement-dependent pathogenic potential of immune complexes containing both ACPA and IgM or IgA RF has been established using autoantibodies from RA patients [78]. Additionally, another ACPA-mediated mechanism of TNF-α induction that may operate in RA has been described. Through binding to surface, over-expressed, citrullinated glucose-regulated protein 78 on RA peripheral blood mononuclear cells, ACPAs selectively activate the ERK1/2 and JNK signaling pathways to enhance IKK-α phosphorylation, which leads to the activation of NF-κB and the production of TNF-α [79].

The pathogenic activity of ACPA in RA is also associated with the induction of neutrophil cellular traps (NETosis), a specific type of cellular death that consists in the extrusion of the intracellular material (DNA, histones, IL-17A, TNF-α, granular and cytoplasmic proteins) by neutrophils. Anti-citrullinated vimentin antibodies were shown to potently induce NET formation [80]. Accelerated NETosis in RA is a source of citrullinated autoantigens and PAD enzymes that can citrulline extracellular proteins when released from intracellular compartments [80,81], further promoting ACPA production. Thus, stimulation of NET formation by ACPA could perpetuate inflammatory and autoimmune processes in RA [74].

In some studies, ACPA specific for mutated citrullinated vimentin (MCV) purified from the serum of RA patients bound to the surface of osteoclasts and osteoclast progenitor cells, induced their differentiation, and activated bone-resorbing activity [82]. All these ACPA-mediated processes may be involved in the development and vicious cycle of RA.

#### 4.3.2. Antigen Presentation

There are three main types of antigen-presenting cells in the human body: dendritic cells (DC), macrophages, and B cells. The presentation of specific antigens via B cell antigen receptor occurs with very high efficiency and results in the activation of cognate T cells [83]. In RA, B cells, such as APCs, present their own antigens to CD4+ T helper cells, primarily. CD4+ helper T cells are divided into follicular helper cells (Tfh) and peripheral helper cells (Tph). The majority of reports revealed that in the blood of RA patients, there is an increase in the circulating Tph and Tfh cell frequency, compared to healthy controls [84]. Pathogenic Th17 cells secrete pro-inflammatory cytokines such as IL-21, which play key roles in synovial inflammation via B cell activation, proliferation, differentiation, affinity maturation, and antibody production [83,85]. The level of serum interleukin (IL)-21 in RA patients showed a positive correlation with the erythrocyte sedimentation rate (ESR), RF, C-reactive protein (CRP), and ACPA [86].

#### 4.3.3. Cytokine Secretion

The synovial membrane of RA patients contains a complex network of cytokines that are related to the development of RA. B cells in the peripheral blood of RA patients can secrete many cytokines including CCL3, TNF-α, IFN-γ, IL-6, IL-1β, IL-17, and IL-18 [23,87]. TNF-α can increase the expression of RANKL by B cells in the presence of IL-1β, thereby promoting the formation of osteoclasts [88,89].

Regulatory B (Breg) cells are a type of B cell that exerts immunosuppressive functions. Breg cells are mainly responsible for the production of anti-inflammatory cytokines (IL-10, TGF-β, and IL-35). Therefore, Breg cells may inhibit RA progression [23,90,91]. Human Breg cells are predominantly enriched in transitional (CD19+CD24hiCD38hi) and memory (CD19+CD24hiCD27+) B cells [23]. RA patients with active disease had reduced numbers of CD19+CD24hiCD38hi B cells in the peripheral blood compared to patients with inactive disease or healthy individuals. These results suggest that CD19+CD24hiCD38hi B cells with a regulatory function may fail to prevent the development of autoimmune responses and inflammation in patients with active RA [92]. Additionally, CD19+CD24hiCD38hi B cells can reduce ACPA production while inhibiting the production of inflammatory factors such as IFN-γ and IL-21 by T cells in RA patients [23].

#### 4.3.4. Osteoclast Activation

Bone osteostasis is regulated by a balance between osteoblastic bone formation and osteoclastic bone resorption. Memory B cells have been described to express NF-κB ligand (RANKL), a key cytokine that regulates bone homeostasis [93,94,95]. Therefore, ACPA-positive RA patients exhibit more pronounced trabecular bone resorption at the distal radius compared to seronegative RA patients, independent of disease duration, activity, and treatments [96].

Some experiments have shown that the increasing number of RANKL-expressing plasma cells enhanced the formation of bone-resorbing osteoclasts. This suggests that plasma cells promote osteoclastogenesis in RA [97].

### 4.4. Innate Immunity-Mediated Immune Response in RA

In recent decades, new research has revealed that the innate immune system plays an important role in the initiation and progression of RA. A variety of innate immune cells, including monocytes, macrophages, and dendritic cells, are involved in the inflammatory response seen in RA patients, and they induce activation of the adaptive immune system, which plays an important role in the later stages of the disease [98]. Macrophages are the most abundant immune cells found in RA synovium, where they produce the predominant pro-inflammatory cytokines involved in RA pathogenesis (TNFα, IL-1β, and IL-6), with chemoattractant factors (CCL2 and IL-8) and metalloproteinases (MMP-3 and MMP-12) [99,100]. Classically activated macrophages (M1) induce joint erosion, secreting mainly pro-inflammatory cytokines such as TNF-α and IL-1. Alternatively, anti-inflammatory cytokines (mainly IL-10 and TGF-β) activate macrophages (M2) which regulate inflammation and contribute to angiogenesis, tissue remodeling, and repair [101]. Recent studies have shown that there is an imbalance of macrophage subsets from the synovial fluid of patients with RA, where the M1/M2 ratio is higher in patients with RA compared to patients with OA [102].

Dendritic cells (DCs) have the ability to induce tolerance or autoimmunity depending on the various signals that they receive from the joint environment [98]. DCs can promote resistance through several mechanisms, including generation and maintenance of Treg cells, as well as induction of T cell unresponsiveness [103]. Conversely, the antigen-presenting ability of DCs may promote priming and/or effector differentiation of self-reactive T cells. In inflamed RA synovium, most antigen-presenting cells (APCs) are fully differentiated DCs expressing high levels of class I and II MHCs and T cell co-stimulatory molecules [98,104]. The phenotype and function of DCs play a complex and dichotomous role in the pathogenesis of RA [98]. There are two major DC subsets involved in the pathogenesis of RA—conventional DCs (cDCs) and plasmacytoid DCs (pDCs) [104]. cDCs can be broadly subdivided into two subsets: cDC1 and cDC2, which are specialized in presenting endogenous and exogenous antigens on both MHC-I and II to CD8 and CD4 T cells. On the other side, pDCs are found circulating in the blood and in peripheral organs and are uniquely able to rapidly produce large amounts of type I interferons upon viral infection [105].

Natural killer (NK) cells may be divided into the CD56dim subset and the CD56bright subset. CD56bright can increase TNF production by CD14+ monocytes in a contact-dependent manner when activated with IL-12, IL-15, or IL-18 [106]. Granzyme B plays a role in promoting autoimmunity, generating new epitopes, and inducing direct cartilage damage [107]. There are three different groups of innate lymphocytes (ILCs): ILC1, ILC2, ILC3. NK cells belong to the first group. ILCs are mostly tissue-resident cells and are deeply integrated into the fabric of tissues, and recent studies have revealed that these cells serve as a bridge between the innate and adaptive immune systems and are characterized by the absence of recombination-activating genes (RAGs)—dependent rearranged antigen-specific receptors [98]. According to a recent study that examined lymph node (LN) biopsy specimens from 12 patients in the earliest phase of RA, no difference in the frequency of total ILC was found, but RA patients had higher numbers of ILC1 and ILC3 in the LNs than seven healthy people. This finding indicates that the ILC distribution in LNs changes from a homeostatic status to a more inflammatory status before and during the early stages of RA development [108]. Additionally, further study has shown that ILC3 CCR6+ cells may play some roles in the development of RA through the production of IL-17 and IL-22 [109]. In contrast to ILC1 and ILC3, ILC2 levels decrease in the synovial membrane of RA patients, whereas levels are higher in the joint/circulatory system when RA patients are in remission (96). Recently, IL-9-producing ILC2 cells have been identified as mediators of molecular and cellular pathways that mediate the resolution of chronic inflammation. In mice, the absence of IL-9 impaired ILC2 proliferation and activation of Treg cells and resulted in chronic arthritis with cartilage destruction. In contrast, treatment with IL-9 promoted ILC2-dependent Treg activation and induced inflammatory resolution. In addition, patients with RA in remission showed high numbers of IL-9 + ILC2 cells in the joints and blood [110].

## 5. Treatments Targeting the Pathogenic Cells and Cytokines

### 5.1. Currently Approved Biologic/Targeted Syntetic DMARDS

TNF-α is a pro-inflammatory cytokine produced when inflammation occurs by macrophages and monocytes [111]. Binding to two distinct receptors (TNFR1 and TNFR2) establishes different signaling cascades that can trigger the apoptosis, differentiation, proliferation, and migration of inflammatory cells [112]. TNF-α inhibitors have been used clinically to counterbalance the high TNF levels accounting for joint inflammation in RA [111]. Currently available TNF inhibitors are etanercept, infliximab, adalimumab, golimumab, and certolizumab pegol. TNF-α inhibitors have markedly improved treatment outcomes in RA [113].

Rituximab is a chimeric monoclonal antibody that targets the CD20 molecule expressed on the surface of 95% of human B cells. Additionally, the use of rituximab to deplete all B cells except pro-B cells and plasma cells is currently the most widely used treatment for RA [23,114].

Abatacept (CTLA4-Ig) has been approved and successfully used to treat RA. Abatacept inhibits the co-stimulation and activation of T cells, leading to the downregulation of inflammatory mediators by binding to CD80 and CD86 on the surface of B cells [115].

IL-6 has been proved to be a central cytokine in RA pathogenesis by contributing to the production of acute-phase proteins involved in the systemic progression of RA. The first IL-6 blocker was tocilizumab, a humanized anti-IL-6 receptor monoclonal antibody. Recently, sirukumab, olokizumab, clazakizumab, and sarilumab have been developed [116,117].

Janus kinase (JAK) mediates signaling through IL-6R and many other transmembrane receptors (cytokine receptors, G protein-coupled receptors, receptor tyrosine kinases) [118,119]. JAK inhibitors are small molecule-targeted therapeutics and are the first oral options to compare favorably with conventional biological disease-modifying antirheumatic drugs (DMARDs); tofacitinib, baricitinib, and upadacitinib are the first JAK inhibitors commercially available [120].

### 5.2. Therapeutic Approaches under Evaluation in Humans

Plasmablasts and plasma cells play very important roles in many autoimmune diseases, such as rheumatoid arthritis (RA). In vitro experiments show that daratumumab (an anti-CD38 monoclonal antibody) ablates plasma cells and plasmablasts in PBMCs of RA patients in a dose-dependent manner. However, the efficacy and safety of daratumumab in the treatment of RA patients still need to be confirmed [121].

Bruton’s tyrosine kinase (BTK), a member of the Tec family of nonreceptor tyrosine kinases, is a cytoplasmic kinase expressed in B cells and myeloid cells [122]. BTK inhibitors are a new class of drugs that inhibit B cell receptor activation, FC-γ receptor signaling, and osteoclast proliferation [123]. Therefore, BTK inhibitors are promising new drugs with potential efficacy in B cell malignancies. Ibrutinib, a first-in-class drug, has become one of the world’s top five best-selling drugs, opening the door to an era of chemotherapy—free management of B cell malignancies [124]. Currently, BTK inhibitors are being investigated to treat a variety of autoimmune diseases, including RA. While the results of BTK inhibitors in animal models of RA are good, the results of subsequent human clinical trials have been somewhat ambiguous [123,125].

IL-21 is a potent pleiotropic cytokine involved in the activation/differentiation of many immune cell types, including B and T cells. IL-21 regulates both innate and acquired immune responses and plays a key role in anti-tumor and antiviral responses, as well as inflammatory responses that promote the onset of autoimmune and inflammatory diseases [126]. Some studies suggested that IL-21 has important biological effects in autoimmunity and might be a promising therapeutic target for RA [127]. A single dose of NNC0114-0005 (≤25 mg/kg IV; ≤4 mg/kg SC), a human recombinant anti-interleukin (IL)-21 monoclonal antibody, was well tolerated in healthy control and RA patients. Accumulation of IL-21-containing complexes suggests neutralization of target cytokines. Based on this, additional experiments were started to investigate the efficacy of anti-IL-21 [128].

Tabalumab, a monoclonal antibody that neutralizes membrane-bound and soluble B cell activating factor (BAFF), was tested in patients with active RA who showed an inadequate response to TNF inhibitors. Although the primary end point was not achieved, an indication of efficacy was observed at earlier time points [129]. Other study results showed that neither clinical efficacy nor significant safety signals were observed with tabalumab, despite evidence of biological activity [130]. More studies are needed.

The CD40/CD40L axis plays a central role in the generation of the humoral immune response and is an attractive target for the treatment of autoimmune diseases in the clinic. VIB4920 (formerly MEDI4920) is an Fc-deficient CD40L antagonist that significantly reduced disease activity, achieving low disease activity or clinical remission in a phase 1 study [131].

### 5.3. Potential Therapeutic Approaches

Autoreactive B cells play an important role in the pathogenesis of rheumatoid arthritis (RA). To address the selective and persistent problem associated with RA therapy, using the universal anti-fluorescein isothiocyanate (FITC) chimeric antigen receptor T cells (CAR-T cells) coupled with FITC-labeled antigenic peptide epitopes, scholars developed a tailored therapeutic strategy that eliminates recognizing B cell subsets. Additionally, the cytotoxicity of the CAR-T cells was dependent on the presence of peptides and occurred in a dose-dependent manner [132].

The therapeutic potential of mesenchymal stem cell transplantation (MSCT) as a treatment for RA is expected. A proliferation-inducing ligand (APRIL), BAFF, and BAFF receptors play important roles in the pathogenesis of RA. MSCT suppressed B cells by decreasing the expression of the BAFF and APRIL genes [133].

Upon exposure to immunogenic stimuli such as microbial pathogens, the initiation of inflammation and immune responses is mediated by Toll-like receptors (TLRs) which activate cells of the innate immune system, including monocytes, macrophages, and dendritic cells [134]. Additionally, TLRs ligate to natural ligands expressed in RA joints, leading to joint inflammation and osteoclast destruction. RA synovial tissue fibroblasts, M1 macrophages, Th16 cells, mature osteoclasts, and endothelial cells play important roles in TLR-mediated RA pathology. Therefore, novel approaches are being tested to target TLR function [135].

## 6. Conclusions

RA is a chronic, systemic autoimmune disease associated with proliferation of joint synovial tissue, formation of pannus, destruction of cartilage, systemic complications, and early death. The basic feature of RA is an autoimmune disorder in which autoreactive CD4+ T cells, pathogenic B cells, macrophages, inflammatory cytokines, chemokines, and autoantibodies are abnormally elevated. Accumulating data show diverse pathogenetic mechanisms of RA involving complex roles of various immune cells. However, much remains to be resolved. Through the advanced understandings of the complex mechanisms of the disease, novel therapies that precisely target pathogenic molecules can be developed.

## Figures and Tables

**Figure 1 ijms-23-00905-f001:**
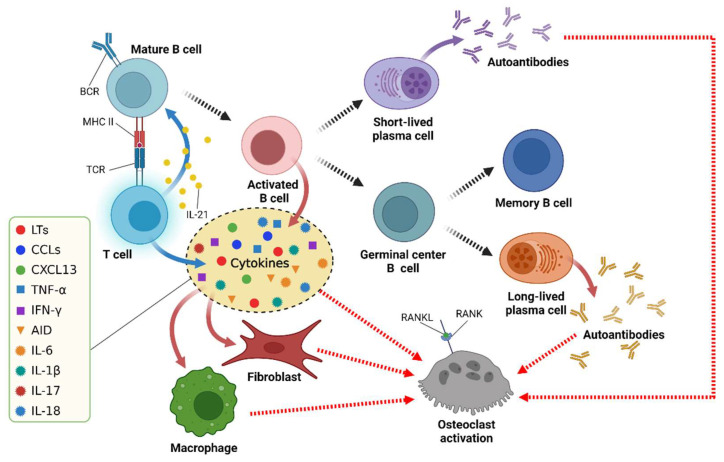
The multiple functions of B cells in RA. BCR = B cell receptor; TCR = T cell receptor; MHC = major histocompatibility complex; RANKL = receptor activator of NF-κB ligand; RANK = receptor activator of NF-κB. Uncited reference: [52]. Created with Biorender.com.

**Figure 2 ijms-23-00905-f002:**
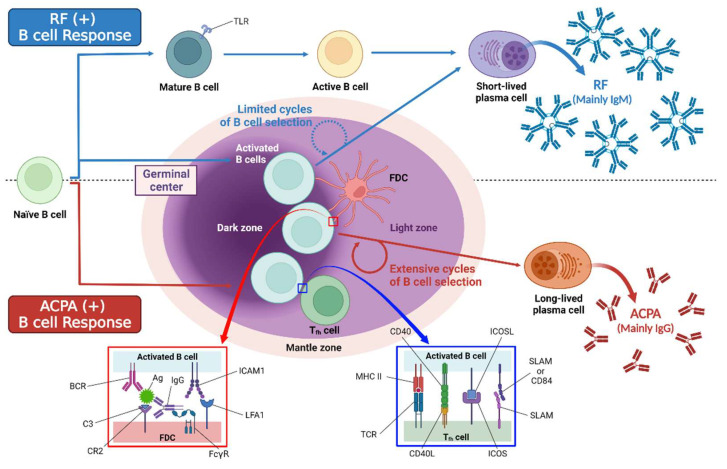
The difference between ACPA-positive B cells and RF-positive B cells. TLR = Toll-like receptor. Uncited reference: [58]. Created with Biorender.com.

**Table 1 ijms-23-00905-t001:** The 2010 ACR/EULAR classification criteria for RA.

Classification Criteria for RA (Total Score ≥ 6 is Considered Satisfactory for the Diagnosis of RA)		Score
joint involvement (swollen or tender joint)	1 large joint (shoulders, elbows, hips, knees, and ankles)	0
	2–10 large joints	1
	1–3 small joints (with or without involvement of large joints) *	2
	4–10 small joints (with or without involvement of large joints)	3
	>10 joints (at least 1 small joint) **	5
serology	Negative RF and negative ACPA (≤upper limit of normal (ULN))	0
	Low-positive RF or low-positive ACPA (≤ULN and ≤3 times)	2
	High-positive RF or high-positive ACPA (≤3 times)	3
acute-phase reactants	Normal CRP and normal ESR	0
	Abnormal CRP or abnormal ESR	1
duration of symptoms	<6 weeks	0
	≥6 weeks	1

CRP = C-reactive protein; ESR = erythrocyte sedimentation rate. * “Small joints” refers to the metacarpophalangeal joints, proximal interphalangeal joints, second through fifth metatarsophalangeal joints, thumb interphalangeal joints, and wrists. ** In this category, at least 1 of the involved joints must be a small joint; the other joints can include any combination of large and additional small joints, as well as other joints not specifically listed elsewhere (temporomandibular, acromioclavicular, sternoclavicular, etc.).

## Data Availability

Not applicable.

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
