# Peer review of "Rheumatoid Arthritis: Pathogenic Roles of Diverse Immune Cells"

_ijms, 2022, doi:10.3390/ijms23020905_

Round 1
Reviewer 1 Report
This review provides updated pathogenesis of Rheumatoid arthritis which will unveil the novel therapeutic targets. This is a very informative report, but I have a few questions before accepting this report.
・Please describe the differences in the pathophysiology of RA between adults and children.
・Please describe the mechanism of the analogous disease of rheumatoid arthritis.
・Please propose a biomarker that could be used in clinical practice.
Author Response
Thank you for your constructive review.
Please see the attachment.

Reviewer 2 Report
I have revised the manuscript entitled “Rheumatoid Arthritis: Pathogenic Roles of Diverse Immune Cells”. In this paper, authors have been performed a review about the implication of diverse immune cells in the autoimmune pathology. The B-lymphocyte implication in RA is the best developed, the rest is a bit sparse. Thus, the review could be very interesting, however there are important issues that are poorly dealt with or missing:
- Ln 34, Authors do not mention virus infections as a possible environmental factor, which is then mentioned in Ln148.
- There is no mention to Th17, Th1 or Treg plasticity. The importance of Th plasticity in RA is great in autoimmune diseases. Th17 cells that change their linage commitment to Th1, and subsets as Th1 exTh17 are develop in RA. The plasticity between Treg and Th17 subset is also important in this pathology.
- There is no mention to Th17 heterogeneity. There are two types of Th17 cells, pathogenic Th17 cells or not pathogenic Th17 cells. In RA, pathogenic Th17 cells play a very important role. Among different cytokines that this Th17 subtype produces is GM-CSF or IL-22. This last cytokine is closely related to osteoclastogenic actions of Th17 cells.
- Ln 116, the article mentioned by the authors (ref. 37) does not talk about the involvement of Th22, but of IL-22. IL-22 is also secreted by pathogenic Th17 cells. It is necessary to measure other factors closely related to Th22 to involucrate this subset in RA, like AHR or perform cytometry studies.
- The involvement of Breg cells is poorly developed, with only one mention in the text.
- There is no mention to the implication of different macrophages subtypes, M1 and M2.
- The involvement of different ILCs in this pathology is poorly addressed and they are innate cells that are becoming increasingly important in this pathology.
- The description of the different treatments in RA is a bit mixed, they do not sort by treatment of groups: NSAIDs, biological DMARDs, synthetic DMARDs. If authors. If the authors prefer to talk in this section only about treatments that target the different cells involved in AR, they should change the title.
Author Response

(The authors gave the same response as above.)

Round 2
Reviewer 2 Report
I have revised the new version of the manuscript entitled “Rheumatoid Arthritis: Pathogenic Roles of Diverse Immune Cells”. The authors have included almost all of my suggestions in this new version, and as a result, the revision is much improved. I consider, therefore, that the review can now be published in IJMS.
Author Response
I really appreciated your amazing and kind advices and feedbacks.
Thank you for being so supportive.